# SARS-CoV-2: Immunity, Challenges with Current Vaccines, and a Novel Perspective on Mucosal Vaccines

**DOI:** 10.3390/vaccines11040849

**Published:** 2023-04-15

**Authors:** Raju Sunagar, Amit Singh, Sudeep Kumar

**Affiliations:** 1Ella Foundation, Hyderabad 500078, India; rajus@ellafoundation.org; 2Department of Immunology and Microbial Disease, Albany Medical College, Albany, NY 12208, USA; singha5@amc.edu

**Keywords:** SARS-CoV-2, COVID-19, mucosal vaccine, intranasal vaccine, APC targeting, human FcγRI, adjuvant, immunity

## Abstract

The global rollout of COVID-19 vaccines has played a critical role in reducing pandemic spread, disease severity, hospitalizations, and deaths. However, the first-generation vaccines failed to block severe acute respiratory syndrome coronavirus 2 (SARS-CoV-2) infection and transmission, partially due to the limited induction of mucosal immunity, leading to the continuous emergence of variants of concern (VOC) and breakthrough infections. To meet the challenges from VOC, limited durability, and lack of mucosal immune response of first-generation vaccines, novel approaches are being investigated. Herein, we have discussed the current knowledge pertaining to natural and vaccine-induced immunity, and the role of the mucosal immune response in controlling SARS-CoV2 infection. We have also presented the current status of the novel approaches aimed at eliciting both mucosal and systemic immunity. Finally, we have presented a novel adjuvant-free approach to elicit effective mucosal immunity against SARS-CoV-2, which lacks the safety concerns associated with live-attenuated vaccine platforms.

## 1. Introduction

The spread of severe acute respiratory syndrome coronavirus 2 (SARS-CoV-2) has led to a global COVID-19 pandemic with a series of SARS-CoV-2 waves. This has resulted in the deployment of an effective vaccine within less than a year from the beginning of the pandemic. Following the completion of 2 years of the COVID-19 pandemic, thus far, more than 180 vaccine candidates are in clinical trials from various manufacturers. Of these, more than 20 vaccines with different platforms have been currently approved globally, and 11 of them have WHO emergency use listing [1] and authorization [2,3]. As of February 2023, 13.3 billion doses have been administered globally leading to 70.81% of the world population receiving at least one dose of a COVID-19 vaccine (https://www.bing.com/covid?vert=vaccineTracker accessed on 5 February 2023). Most of these vaccines are designed to elicit an immune response against the spike protein that is critical for SARS-CoV-2 binding and cell entry. Similar to SARS-CoV-2 infection, COVID-19 vaccines elicit early production of humoral immune responses and also stimulate long-lasting memory B- and T-cell responses [4,5]. Subsequently, clinical trials and real-world data showed COVID-19 vaccines are highly effective against SARS-CoV-2 symptomatic diseases [6]. Thus, the global rollout of COVID-19 vaccines has played a critical role in reducing pandemic spread, disease severity, hospitalizations, and deaths [7]. More recently, the world has seen a plateau in severe infections, hospitalizations, and deaths which has been largely attributed to the rapid development and deployment of COVID-19 vaccines [8].

## 2. Immunity to SARS-CoV-2

### 2.1. Humoral Immunity

Neutralizing antibodies (nAbs) have emerged as a crucial predictor of survival in COVID-19 patients, with their presence providing robust protection against subsequent reinfection with the same strain [9]. To better understand the role of nAbs, researchers have employed ACE2 receptor inhibition assays and pseudo-virus neutralization assays that utilize pseudo-typed viruses transfected with SARS-CoV-2 spike protein. Of particular interest are Abs against epitopes in S1, which give the most accurate correlates of protection against SARS-CoV-2 infection. Both neutralization and binding assays have considerable prognostic value, wherein binding Ab titers show the highest statistical correlation with protection. Studies suggest that the protective functions of Abs are not limited to neutralization alone, as antibodies with multiple functions can also influence efficacy. Additionally, cellular responses, particularly those involving CD4^+^ and CD8^+^ T cells, also play a critical role in controlling viral replication and augmenting the effects of antibodies. Although there are excellent reviews regarding the humoral immunity to SARS-CoV-2 [10,11,12,13,14], herein we discuss briefly the humoral immunity generated in response to infection and vaccination and its role in controlling COVID-19.

#### 2.1.1. Neutralizing Antibodies

After being infected with SARS-CoV-2, most people develop nAbs relatively quickly. These Abs are produced by B cells using a wide range of heavy and light chain genes of antibodies, and exhibit little to no somatic hypermutation [15,16,17]. This means that the development of nAbs is generated by many B cells with little affinity maturation. Furthermore, nAb responses generally develop from naive B cells, rather than preformed cross-reactive memory B cells [6,18,19,20,21]. The highly immunogenic and easily recognizable receptor-binding domain (RBD) on the SARS-CoV-2 virus is the prime target of nAbs. However, it is important to note that a significant proportion of people who have recovered from COVID-19 have relatively low circulating SARS-CoV-2 nAb titers, indicating that either the potency or serum concentration of the neutralizing antibodies is suboptimal in these individuals [15,22,23].

#### 2.1.2. Natural Infection Ab Response

Following SARS-CoV-2 infection, the vast majority of individuals develop an Ab response within 5–15 days [8,24,25,26,27]. This response targets a variety of antigens, including spike and nucleocapsid proteins [28]. Among these, Abs against the RBD [1] of the spike are the most common, comprising over 90% of nAbs against COVID-19 [1,8,15,16,28], while some nAbs have been shown to target the N-terminal domain of the spike [29]. Notably, the spike protein is highly immunogenic, with spike-specific Abs found in 91–99% of cases [23,30]. In infected individuals, spike IgG, IgA, and IgM Abs all develop simultaneously [8,26,31].

#### 2.1.3. Vaccine-Induced Ab Response

Most COVID-19 vaccines have been designed to elicit immune responses, specifically nAbs [32], against the spike protein of the SARS-CoV-2 virus [33]. Several types of vaccines, such as mRNA, adenoviral-vectored, protein subunit, and whole-cell inactivated virus vaccines, have reported efficacy in phase III trials and have received emergency approval in many countries [33,34]. Recent studies have shown that the mRNA vaccines are effective even after just one dose, with detectable non-nAbs and moderate Th1 cell responses, but almost no nAbs [35]. Interestingly two doses of mRNA vaccine induced significantly higher nAb response compared to convalescent serum [36]. In contrast, the adenovirus vaccines elicit polyfunctional Abs that can neutralize the virus and drive other Ab-dependent effector functions, as well as potent T-cell responses, after a single dose [37,38]. These findings suggest that protection may not solely rely on nAbs but also involve other immune effector mechanisms, including non-nAbs, T cells, and innate immune mechanisms.

#### 2.1.4. Mechanism of Ab Induction

In addition to antigen concentration, ab response also depends on cellular pathways. Antigen concentration has been found to correlate with Ab titers in various preclinical studies and appears to be true for SARS-CoV-2 as well. Studies have shown that higher nAb titers and total spike Ab titers correlate with COVID-19 disease severity [15,28], similar to what was observed in SARS and MERS. At this point, the relationship between nAbs, T_FH_ cells, and SARS-CoV2 infection is complex. High nAb titers correlate with severe disease and extrafollicular B cell responses, while SARS-CoV-2-specific T_FH_ responses exhibit differential associations in various studies. The disconnect between B-cell and T-cell responses and disparate T- and B-cell responses may be attributed to variations in early innate immune responses, resulting in delayed kinetics or dysregulated T-cell priming [39,40]. To fill the knowledge gaps, more longitudinal studies are required to understand the dynamics of virus-specific Ab response and T-cell responses during acute SARS-CoV-2 infections exhibiting varying degrees of disease severity.

#### 2.1.5. Mechanism of Ab-Mediated Protection

While it is known that Abs typically prevent viruses from infecting cells, they can also induce the killing of virally infected cells (Figure 1), which is an important mechanism of action in vivo [41,42,43]. Although currently there is no direct evidence of this occurring in human SARS-CoV-2 infections, studies with various animal models suggest the protective role of Fc receptor-associated Ab functions against SARS-CoV-2. SARS-CoV-2 specific Abs have been found to exhibit NK cell-mediated Ab-dependent cell-mediated cytotoxicity (ADCC) [43,44]. Complements function by opsonization and Ab-dependent complement deposition (ADCD) of the virally infected cells, whereby the viral glycoproteins expressed on the cell surfaces recognized by Abs serve as a platform for complement activation and deposition [43,45]. Ab-dependent cellular phagocytosis (ADCP), is another Fc-dependent mechanism, which has been thought to play a protective role against SARS-CoV-2 [43]. Additionally, studies in mice have shown that nAbs with Fc-receptor binding capacity are more protective [43,46], while patients who succumbed to the COVID-19 infection exhibit reduced Fc-dependent Ab effector response [47].

### 2.2. Cell-Mediated Immunity

Although nAbs were initially considered to be most important in protection against COVID-19 [48,49], recent evidence suggests that virus-specific T cells may synergize with nAbs for robust protection against SARS-CoV-2 [9]. The role of cell-mediated immunity (CMI) in controlling SARS-CoV-2 infection was recognized in the clinical trials with monoclonal Abs (mAbs), wherein mAb treatment clearly provided clinical benefit in the early stage COVID infection. Even though the likelihood of hospitalization was significantly reduced [50], the mAb treatment promoted only a modest reduction in viral load. On the contrary, individuals who are immunized/infected exhibited a substantial reduction in viral load [51]. This suggests that CMI may be critical in minimizing viral burden by controlling and eliminating infected cells [9]. Moreover, several clinical trials revealed that high-dose neutralizing mAb has no beneficial effect in hospitalized individuals. However, the administration of mAbs before infection has been highly effective in preventing severe disease following SARS-CoV2 infection in humans as well as non-human primates [9,52,53]. Thus, while early mAb treatment provides clinical benefit to high-risk individuals at early stages of COVID/before infection, the actual impact of mAbs on viral loads is only modest, suggesting that Abs may not be the major effector to viral clearance and that mAb treatment can mostly restrict viral pathogenesis before the patients mount a robust CMI response and clear the virus [54].

#### 2.2.1. T-Cell Response Induced by SARS-CoV-2 Infection

Demonstrating the protective roles of T cells in humans is a difficult task. Measuring antigen-specific T cells is more technically challenging and expensive than measuring Ab responses. Since T-cell receptors do not bind to antigens directly, quantification of antigen-specific T-cell response relies on either MHC- peptide tetramers or in vitro stimulation-based T-cell response. Moreover, unlike Abs, a simple adoptive transfer approach cannot be applied to demonstrate the role of T cells in immunity to SARS-CoV-2. However, indirect evidence of the protective roles of T cells comes from studies involving mAb therapy. Even with very high levels of nAbs, viral loads were only reduced by fourfold, while individuals who developed immune responses on their own exhibited a 1000-fold greater reduction in viral loads [50]. Despite not being directly measured in clinical trials, CD4 T-cell responses specific to spike as well as other SARS-CoV-2 proteins [22,55,56], composed of Th1 cells, T_FH_ cells, and CD4-CTL cells [55,57,58], develop in nearly 100% of cases. The vast majority of infected individuals also develop memory Th1 and T_FH_ cells [22], while CD8 T-cell responses, recognizing multiple SARS-CoV-2 proteins [55,56], are detectable in around 70% of individuals [25,55], and CD8 T-cell memory is detectable in the majority of individuals [22,59]. In conclusion, CD4 and CD8 T-cell responses are generated in most human SARS-CoV-2 infections.

Several studies have examined the relationship between T cells and the control of SARS-CoV-2 infection. One such study found that individuals with strong early T-cell responses had a milder disease and faster viral clearance, while those with few virus-specific T cells were associated with severe COVID-19 along with sustained high viral loads [60]. However, this study had limitations, including a relatively small number of individuals and the inability to distinguish between CD4 and CD8 T cells [60]. Another study examined SARS-CoV-2-specific CD8 T cells, CD4 T cells, and nAbs in 52 individuals and observed a correlation between the presence of T-cell response and limited disease severity, but no correlation with Ab response [25]. Age was also found to be correlated with weaker T-cell responses, indicating a link between age and COVID-19 severity [25]. Moreover, other independent studies also found diminished SARS-CoV-2-specific CD8 T-cell responses in hospitalized patients, while nucleoprotein-specific CD8 T cells were associated with stronger responses and milder disease [61,62,63]. Overall, these studies indicate important roles for T cells in the control and clearance of SARS-CoV-2, with larger and earlier T-cell responses associated with better clinical outcomes.

Epidemiological studies in patient populations with compromised Ab responses, such as individuals with agammaglobulinemia B-cell depletion, exhibit only a moderate increase in hospitalization rates, suggesting that CMI can effectively control and clear SARS-CoV-2 without a significant Ab response [64,65,66,67,68,69]. However, limitations of these studies include the lack of direct measurement of T-cell responses in these individuals. Animal models of COVID-19 have been challenging to study in terms of T-cell efficacy due to their rapid disease progression, which is typically faster than in humans. Autopsy and bronchoalveolar lavage samples from COVID-19 patients have consistently shown a correlation between reduced T-cell presence and severe disease, providing further evidence for the important role of T cells in the control and clearance of SARS-CoV-2 [70,71,72,73,74,75]. Notably, depletion of CD8 T cells after SARS-CoV-2 infection resulted in significantly increased viral burden upon rechallenge of the animals with SARS-CoV-2 [76]. Moreover, following the successful resolution of mild COVID-19, tissue-resident memory CD4 T cells and CD8 T cells specific for SARS-CoV-2 have been found in the lungs of humans, indicating that virus-specific T cells do migrate to the lungs during the course of the disease [77]. Overall, these lines of evidence strongly support the crucial role of T cells in the prevention of severe COVID-19.

#### 2.2.2. T-Cell Response Induced by COVID-19 Vaccines

The mRNA COVID-19 vaccines, Pfizer BNT162b2 and Moderna mRNA-1273, induce Th1 [78] and T_FH_ cells and generate memory CD4 T cells [79] that persist for at least six months post-vaccination [80]. Despite initial confusion, it has now become clear that most individuals also develop spike-specific CD8 T cells in response to these vaccines [4,81,82,83]. T cells play a critical role in generating protective immunity to mRNA COVID-19 vaccines by providing T_FH_-cell help to B cells for the production of affinity matured nAbs and memory B cells [79,84,85,86,87].

The Pfizer BNT162b2 and Moderna mRNA-1273 COVID-19 vaccines have been shown to provide significant protective efficacy against the ancestral strain and the Alpha variant after a single dose, despite low or undetectable levels of nAbs [88,89]. This indicates that T cells may play a role in protection against COVID-19. In non-human primates, CD4 T cells have been identified as a correlate of protection against SARS-CoV-2 for the mRNA-1273 vaccine. Moreover, spike-specific CD4 T cells expressing CD40L, IL21, or any Th1 cytokine correlated with significantly reduced viral loads [83,90]. Importantly, these CD4 T cells were still associated with protective immunity even when considering the contributions of spike-specific Ab responses in multivariate analysis [90]. However, it should be noted that evaluation of T cell-mediated protective immunity is challenging in the rhesus monkey model, and limitations of the study include the inability to independently demonstrate the role of T cells and the largely undetected spike-specific CD8 T-cell responses after mRNA-1273 immunization. Sampling peripheral blood and quantification of antigen-specific T-cell responses does not provide direct evidence of the protective role of T cells. Other studies have also shown that CD4 T-cell responses correlated with protection, and CD8 T-cell responses exhibited an even stronger correlation with protection [91]. A SARS-CoV-2 intranasal vaccine directed to induce T-cell responses has also demonstrated protection in rhesus monkeys without detectable nAbs [92].

The type of COVID-19 vaccine can impact T-cell responses. Adenoviral COVID-19 vaccines, such as AstraZeneca ChAdOx1 and J&J Ad26.COV2.S, elicit Th1, T_FH_, and CD8 T-cell responses [93]. Inactivated virus COVID-19 vaccines, such as CoronaVac and Covaxin (BBV152), generate relatively weak CD4 T-cell responses and a mixture of Th1 and Th2 cells, but Covaxin may be more effective due to substantial Th1 and T_FH_-cell responses [94]. Two-dose mRNA vaccine protection against detectable infection appears to wane over 6 months, but protection against hospitalizations and deaths remains relatively stable. This suggests a role for vaccine-elicited T cells in protective immunity, which may also be significant for Omicron, as individuals with no detectable Omicron nAbs still have protection from hospitalizations or fatalities. However, further research is needed to fully understand the role of T cells in vaccine-induced immunity over time and compared to different vaccine types.

#### 2.2.3. T-Cell Mechanisms of Protection

The following section introduces various cellular pathways that contribute to immunity against SARS-CoV2. T cells play a crucial role in protective immunity against infectious diseases through various mechanisms (Figure 2). CD8 T cells directly recognize and kill infected cells, making them important in many viral infections (Figure 2) [95]. In fact, in viral infections, CD8 T cells are essential for fully eradicating the virus [96]. CD4 T cells contribute to protective immunity against viruses by at least three distinct mechanisms (Figure 2). The first is mediated by T_FH_ cells that help B cells generate nAbs. Weak or no nAbs are produced in the absence of T_FH_ cells, and long-term Ab production, which comes from long-lived plasma cells, also depends on the presence of these cells [96,97]. The second mechanism involves Th1 cells that produce cytokines, such as interferon-γ, to enhance the antiviral state of infected cells and recruit other effector cells to the site of infection. Th1 cells have been found to play important role in protective immunity against influenza and SARS-CoV-2 [98,99]. The third mechanism involves CD4-CTL cells that express granzymes and have cytotoxic activity similar to CD8-CTL cells. CD4-CTL cells have been observed in multiple viral infections and are associated with protection against Dengue and influenza in humans [100,101,102,103,104]. The location of T cells is also critical to their function, with tissue-resident memory T cells being an essential aspect of T-cell biology. These cells play an important role in providing protection against reinfection by remaining in tissues that are more susceptible to infection.

## 3. Challenges of the Current Vaccine Approaches

Waning immunity coupled with the immune escape ability of the Omicron variant is causing a high prevalence of reinfection and vaccine breakthrough infections among COVID-19 vaccinees. To control the pandemic, it is essential to suppress community transmission of the virus. While public health interventions have an important part to play, immunological control of transmission will require the induction of antiviral Abs in respiratory and oral secretions, which are the source of the infective droplets and aerosols. A key question in this context is why systemically immunized subjects continue to have the virus in their salivary and nasal secretions.

Breakthrough infections in fully vaccinated individuals may occur due to a waning immune response and poor or absent mucosal immunity, which is critical for blocking SARS-CoV-2 infection and transmission. One potential cause of breakthrough infections is the emergence of new variants of concern (VOC) that escape immunity, thereby reducing the effectiveness of the vaccine. The emergence of SARS-CoV-2 VOC has raised concerns about the breadth and durability of nAb responses [105]. The following subsections present some of the challenges associated with the current vaccine approaches.

### 3.1. Persistence/Waning of SARS-CoV-2 Vaccine-Induced Immunity

Once a vaccine is licensed and introduced into the population, it is important to conduct post-licensure studies to measure vaccine effectiveness (VE), as vaccine performance in a real-world setting can differ from efficacy measured under controlled trial conditions [106]. In the case of COVID-19 vaccines, studies following vaccine rollout have shown them to be highly effective at preventing COVID-19 illness, including severe disease [107], as shown in Table 1. However, despite the high effectiveness of current COVID-19 vaccines against serious illness, hospitalization, and death, health care records and immunological studies have revealed a decline in nAbs and vaccine response kinetics among fully vaccinated individuals. This has led to breakthrough infections, particularly among immunocompromised individuals or those with high exposure to the virus [108,109].

Studies have shown that mRNA vaccines induced immunity and protection against SARS-CoV-2 infection, but immunity appeared to wane 3 months after the second dose, particularly in infection-naïve individuals with the BNT162b2 vaccine [109]. However, there was no evidence of declining effectiveness over time against severe, critical, or fatal COVID-19 [110,111]. Retrospective studies in Brazil and Scotland revealed waning vaccine protection of ChAdOx1 nCoV-19 against COVID-19 hospital admissions and deaths within three months of the second vaccine dose. Similarly, the effectiveness of the Ad26.COV2.S vaccine decreased from 74.8% at 1 month to 59.4% at 5 months. A prospective study of BNT162b2- and CoronaVac-vaccinees in Hong Kong demonstrated lower nAb titers and waning nAb response among CoronaVac-vaccinees, indicating a higher risk of pandemic variant breakthrough infection [112,113].

While the persistence of Ab and T-cell responses to the Sinopharm/BBIBP-CorV vaccine showed a decline in Ab responses over time, T-cell responses persisted in all age groups [114,115]. A study of BBV152 six months after the second dose found that more than 75% of participants still had detectable nAbs and cell-mediated immunity to both homologous and heterologous strains, although the magnitude of the responses had declined [116,117]. Finally, recipients of COVAXIN and Covishield have displayed reduced neutralization activity against the Omicron variant, while vaccinees who had a natural infection during the Delta variant-led surge had better neutralization titers for the Omicron variant. Overall, vaccine-induced immunity against SARS-CoV-2 infection wanes over time, even though protection against reducing the risk of SARS-CoV-2 hospitalization and death persists at a robust level for 6 months after the second dose [118,119].

**Table 1 vaccines-11-00849-t001:** Comparison Of WHO EUA Qualified COVID-19 Vaccines.

Platform	Vaccine	Manufacturer	Dosage Number and Schedule	Efficacy against Symptomatic COVID-19 (Clinical Trials)	Adjusted Vaccine Effectiveness (95% CI)
Nucleic acid vaccines	COMIRNATY (BNT162b2)	Pfizer	Two doses, 21 days apart	95 (90.3–97.6) 63	94 (90–96) [51]
	SPIKEVAX (mRNA-1273)	Moderna	Two doses, 28 days apart	94.1 (89.3–96.8) 65	93.5 (91.9–94.7) [60]
Viral vector vaccine	VAXZEVRIA (ChAdOx1-nCoV-19)	AstraZeneca	Two doses, 12 weeks apart *	74.0 (65.3–80.5) 55	63.1 (51.5–72.1) [120]
	Covishield	Serum Institute of India	Two doses, 12 weeks apart	NR	80 (73–86) [121]
	Ad26CoV2.S	Johnson & Johnson	Single dose	66.9 (59.0–73.4) 68	73.6 (65.9–79.9) [122]
	CONVIDECIA (Ad5-nCoV)	CanSino Biologics Inc.		57.5 (39.7–70.0) 70	NR
Inactivated	Corona Vac	Sinovac	Two doses, 14 days apart	83.5 (65.4–92.1)-Turkey71 65.3 (20.0–85.1)-Indonesia72 50.7 (36.0–62.0)-Brazil73	59.0 (43.7–70.2)-Brazil [123] 65.9 (65.2–66.6)-Chile [124]
	BBIBP-CorV	Sinopharm	Two doses, 21 days apart	78.1 (64.8–86.3) 76	79.8 (78–81.4)
	COVAXIN (BBV152)	Bharat Biotech	Two doses, 28 days apart	77.8 (65.2–86.4) 77	50 (33–62) [125] 69 (54–79) [121]
Recombinant	NUVAXOVID (Nvx-CoV-2373)	Novavax	Two doses, 21 days apart	89.7 (80.2–94.6) 79	(19.9–80.1) [126]
	Covovax	Serum Institute of India		NR	NR

* Vaccine was administered at 4 weeks apart in some efficacy trials; NR—Not reported.

### 3.2. Vaccines against Variants

Studies have shown that vaccines such as ChAdOx1nCoV-19, BBV152, and Covishield have a reduction in nAb titers against Alpha and Delta variants compared to the wild-type SARS-CoV-2. However, antigen-specific CD4 and CD8 T-cell responses were conserved against Omicron, Delta variants, and wild-type SARS-CoV-2 [127,128]. Although mRNA vaccines have shown no decline in vaccine effectiveness against Alpha and Beta/Gamma SARS-CoV-2 variants, their effectiveness against the Omicron variant has dropped to 65%. The reduction in neutralizing activity against variants is reflected in the vaccine’s effectiveness. The mRNA vaccination is highly effective in preventing COVID-19-associated hospital admissions related to the Alpha, Delta, and Omicron variants [129], but a booster vaccine dose is required to achieve protection against the Omicron variant similar to the protection that two doses provided against the Delta and Alpha variants [130,131].

AstraZeneca (AZD1222), also known as the Vaxzevria and Covishield vaccine, is effective against severe disease with Alpha, Beta, and Delta strains but shows significantly reduced efficacy against the Omicron variant, although the nAbs can be enhanced with boosters [130,132,133,134]. Janssen (JNJ-78436735; Ad26.COV2.S) COVID vaccine exhibits about 74% to 90% protection against severe morbidity with Alpha, Beta, and Delta variants, but with the Omicron variant, there are significant breakthrough infections and hospitalizations among the recipients of this vaccine although protection against severe disease can be achieved with booster immunizations [135,136].

BBIBP-CorV/NVSI-06–07 retained nAbs against the Alpha, Beta, and Delta variants; however, it drastically declined against the Omicron variant [137,138]. Moderna COVID-19 Vaccine (mRNA-1273) exhibited more than 90% protection against severe illness and hospitalization against the Alpha, Beta, and Delta variants; however, the nAb titer is 35 times lower against the Omicron variant although optimal nAb titers can be restored with a booster dose [136,139].

Similarly, Sputnik V, CoronaVac, and Covaxin (BBV152) exhibited a high degree of efficacy against severe disease with Alpha, Beta, and Delta variants but were less protective against the Omicron variant. In most cases, nAb titers against the Omicron were significantly increased following a booster with the same or a different vaccine [136,140].

### 3.3. Mucosal Immunity Induced by Parenteral Vaccination

Current vaccines were designed for intramuscular delivery with the aim to elicit spike-specific nAb response in the blood. While IM-delivered vaccines did induce robust nAbs against their target antigens in the circulation and prevented vaccinated individuals from severe disease following infection, the rate of breakthrough infections in vaccinated individuals remained high at the time of the peak COVID-19 pandemic. This implies that systemic immunization is inadequate at controlling transmission and as a consequence community spread, although this fact is complicated by the emergence of antigenically diverse variants of SARS-CoV-2.

Immunization with mRNA vaccines via the IM route induces robust humoral and cellular immunity in the circulation. However, vaccinated individuals develop significantly lower nAb against variants including Delta (B.1.617.2) and Omicron BA.1.1 in the respiratory mucosa in comparison with COVID-19 convalescents, even in presence of robust spike-specific Ab responses in the blood. Furthermore, mRNA vaccination induces circulating spike-specific B and T cell-mediated immunity, but in contrast to COVID-19 convalescents, these responses were absent in the respiratory mucosa of vaccinated individuals. In a mouse immunization model, Tang et al. demonstrated that systemic mRNA vaccination alone induces weak respiratory mucosal nAb responses. However, a combination of systemic mRNA vaccination and a mucosal adenovirus-S booster dose induces strong nAb responses not only against the ancestral virus but also the Omicron BA.1.1 variant [141]. Similarly, other reports also suggest that IM immunization does not elicit effective mucosal immunity, especially against variants [142,143,144,145,146].

## 4. Perspective on Mucosal Vaccine Strategies

### 4.1. Role of Mucosal Immune Response in Immunity against SARS-CoV-2

Despite receiving intramuscular SARS-CoV-2 mRNA vaccines, many people still experience breakthrough infections of the Omicron subvariant [147]. While T and B cells induced from the vaccine do provide some protection against the immune evasion of Omicron, it is unclear why there are so many breakthrough infections. Estimated protection (95% CI) against Omicron infection was consistently significantly higher among vaccinated individuals with prior infection compared with vaccinated infection-naïve individuals, with 65% (63–67%) vs. 20% (16–24%) for one dose, 68% (67–70%) vs. 42% (41–44%) for two doses, and 83% (81–84%) vs. 73% (72–73%) for three doses. For individuals with prior infection, estimated protection (95% CI) against Omicron-associated hospitalization was 81% (66–89%) [148]. The reason could be that natural infection elicits stronger humoral immunity in the mucosal surface compared to mRNA vaccination. The reduced immunity by mRNA vaccination to variants was associated with diminished SARS-CoV-2-specific B- and T-cell memory in the respiratory mucosa [148].

On the other hand, compared to a systemic mRNA booster, a mucosal Ad5-S booster immunization can elicit broadened Ab neutralization against VOCs in the bronchoalveolar lavage (BAL). COVID-19 vaccines that are currently approved for emergency use are administered through IM injection and predominantly elicit IgG while offering limited mucosal protection, even though SARS-CoV-2 is transmitted through infectious respiratory fluids [141,144,146,149]. This lack of mucosal immunity could be a contributing factor to the emergence of VOCs and breakthrough infections, indicating the risk of transmission from vaccinated individuals. Therefore, it is essential to develop effective vaccination strategies that include mucosal vaccines and induce front-line mucosal immunity [150]. The inclusion of mucosal boost strategies may offer promise in blocking the transmission and spread of the virus, which can have a significant impact on reducing the extent and spread of the pandemic [149].

Previous studies have demonstrated that IN delivery of influenza vaccines can generate sterilizing immunity and prevent the transmission of the influenza A virus. More recently, it has been found that a single IN dose of chimpanzee adenovirus-based SARS-CoV-2 vaccine (ChAd36-SARS-CoV-2-S) confers superior immunity against SARS-CoV-2 challenge compared to one or two IM immunizations of the same vaccine [124,125]. IN vaccination has also been shown to prevent upper and lower respiratory tract infection and inflammation by SARS-CoV-2 in highly susceptible K18-hACE2 transgenic mice and Syrian golden hamsters, as well as rhesus macaques [125]. This stands in contrast to IM delivery of ChAdOx1 nCoV-19 in rhesus macaques, where immune animals only cleared the SARS-CoV-2 viral load in the lower respiratory tract [149]. Mucosal vaccination, especially through IN delivery of recombinant adenovirus-vectored vaccines, provides several advantages over conventional IM vaccination, particularly against respiratory diseases. The IN route mimics the route of natural infection, leading to a robust immune response. In contrast to parenteral administration, IN immunization of recombinant adenovirus-based vaccines has circumvented pre-existing immunity and conferred sufficient protection against challenge with a variety of pathogens [124,125].

In order to effectively combat an infection, the immune system must respond directly at the site of the infection. However, most assessments of human adaptive immunity are conducted through blood samples, due to their convenience. It is important to note that immune cells and Abs in the blood may not accurately reflect the immune response present in the infected tissue [151]. To better understand the relationship between the immune response in blood and tissues, it is crucial to directly measure immune responses in affected tissues whenever possible, although such data are currently limited for COVID-19 patients. Despite this, recent data suggest that blood measurements largely reflect the local immune response. IgG and IgA Abs are secreted in mucosal tissues, and it is reassuring to note that most individuals display positive levels of spike IgG and IgA in their blood, with these levels correlating with levels found in saliva [31]. Additionally, Anti-RBD secretory IgA has been shown to have enhanced neutralization potency against SARS-CoV-2, as secretory IgA is dimeric [152].

Severe COVID-19 has been associated with higher T-cell frequencies in circulation, However, early studies suggest that SARS-CoV-2-specific T cells may not be found in the blood, as they may have relocated to the lungs. However, the presence of T cells in BAL was associated with better survival and younger age, which is largely consistent with the association of T cells with protection rather than immunopathology [153].

### 4.2. Mucosal Vaccines

Data from preclinical trials suggest that the mucosal vaccines elicit not only systemic immune responses but mucosal immune responses as well, which is equally or more potent than the systemic immunization-elicited immunity in controlling SARS-CoV-2 infection. A very important feature of IN-delivered mucosal vaccines is that they also induce cross-protection against variants including the Alpha, Beta, Gamma, and Omicron variants [154,155,156,157]. IN immunization-induced mucosal immunity is associated with secretory IgA [154,158] and CD4 and CD8 T cell-mediated immune response. With regard to immunity against SARS-CoV-2, viral neutralization by the secretory Abs is considered to be the key mechanism. Evidence suggests that T cell-mediated immunity also contributes significantly [154,156,157], and in some cases, T cell-mediated immunity alone in absence of a spike-specific Ab response is sufficient in controlling the SARS-CoV-2 challenge [92]. In addition, preliminary evidence indicates that IN immunization is also effective in transmission blocking [159,160], which may have a far greater impact on the rapid containment of pandemics. In this regard, however, more investigation is needed.

#### 4.2.1. Preclinical Trials

Numerous studies have investigated the effectiveness of mucosal vaccines in combating the COVID-19 pandemic, with promising results in animal models. One notable study by Du et al. used a preparation of the recombinant receptor-binding domain from SARS-CoV-2 combined with an adjuvant, aluminum oxyhydroxide gel (Alhydrogel R), to immunize mice. The study compared three routes of vaccine administration—intranasal, subcutaneous, and intramuscular—and found that intranasal immunization resulted in the strongest mucosal and humoral responses, with significant amounts of sIgA secreted by B cells from the nasal cavity and lung mucosa. This suggests that intranasal administration of the vaccine may provide the first line of defense against the virus and prevent it from infecting cells [158].

Americo JL et al. compared IM and IN administration of a live, replication-deficient modified vaccinia virus Ankara (MVA)-based SARS-CoV-2 spike vaccine to evaluate protective immune responses in an hACE2-expressing mouse model. IM vaccination-induced spike-specific IgG and nAbs in the lungs, whereas intranasal vaccination also induced IgA and higher levels of antigen-specific CD3+CD8+IFN-γ+ T cells. Similarly, IgG and nAbs were present in the blood of mice immunized with IN and IM, but IgA was detected only after intranasal inoculation. Furthermore, IN boosting increased IgA after IN/IM priming. While IM vaccination prevented morbidity and cleared SARS-CoV-2 from the respiratory tract within several days after challenge, IN vaccination was more effective, as neither an infectious virus nor viral mRNAs were detected in the nasal passage or lungs as early as 2d after challenge, indicating prevention or rapid elimination of SARS-CoV-2 infection. Additionally, nAbs persisted for more than 6 months and that serum, induced to the Wuhan spike protein, neutralized pseudoviruses expressing the spike proteins of variants, although with less potency, particularly for Beta and Omicron [157].

In another study by Hassan et al., a chimpanzee adenovirus vector encoding the spike protein of SARS-CoV-2 was administered to an hACE2-expressing mouse model. A single IN dose of the vaccine induced a robust systemic as well as mucosal immune response including high levels of neutralizing sIgA Abs. This vaccine approach provided almost complete protection against SARS-CoV-2 infection in the upper and lower respiratory tract. In contrast, when the same vaccine was administered via the IM route, no mucosal response was induced, and a significant viral load was observed in the lungs [124].

Wu et al. reported the high effectiveness of IN immunization against SARS-CoV-2 in two animal models. Their study demonstrated that mice immunized intranasally with a replication-defective human type 5 adenovirus that encodes the SARS-CoV-2 spike protein (Ad5-nCoV) were completely protected against upper and lower respiratory tract infection after a single dose of the vaccine. Similarly, in ferrets, one intranasal dose of Ad5-nCoV was sufficient to build immunity in the upper respiratory tract. The study showed that the nasally administered preparation significantly reduced the level of virus replication in the upper respiratory tract, indicating its potential to prevent viral transmission [160].

An et al. developed a vaccine preparation using parainfluenza virus type 5 expressing the SARS-CoV-2 spike protein (CVXGA1) and tested its efficacy in animal models. In mice expressing human ACE-2 receptor, CVXGA1 immunization completely protected against the SARS-oV-2 in lung and brain tissues, as well as significantly reducing viral pneumonia. Furthermore, in the ferret model, IN immunization of CVXGA1 generated high titers of Abs, reduced viral replication I the upper respiratory tracts, and blocked contact transmission to uninfected animals [159].

Wong et al. constructed a two-dose vaccine (BReC-CoV-2) by combining the RBD antigen with Diphtheria toxoid (EcoCRM^®^). The vaccine incorporates Bacterial Enzymatic Combinatorial Chemistry (BECC), BECC470, as an adjuvant. IN administration of BreC-CoV-2 in hACE2 mice induced a strong systemic and mucosal immune response that provided protection against the Washington strain of SARS-CoV-2. IN-induced protection was associated with significantly reduced viral load in the lung, robust RBD-specific IgA titers in the BAL, and induction of broad nAbs in the serum. BReC-CoV-2 immunization using an IM prime and IN boost approach protected mice from a lethal challenge of the Delta variant of SARS-CoV-2. IN administration of BReC-CoV-2 induced better protection than IM-only administration to mice against the lethal challenge of SARS-CoV-2 [156].

Bharat Biotech conducted a preclinical evaluation of ChAd-SARS-CoV-2-S (BBV154), a replication-defective chimpanzee adenovirus (ChAd)- based COVID-19 vaccine in mice, rats, hamsters, and rabbits. BBV154 encodes a prefusion-stabilized version of the SARS-CoV-2 spike protein with two proline substitutions in the S2 subunit. Repeated dose toxicity studies exhibited excellent safety profiles in terms of pathology and biochemical analysis. IN administration of BBV154 elicited robust mucosal and systemic humoral immune responses as well as a Th1-biased CMI response. BBV154 IN vaccination also elicited a potent nAb response against the Omicron variant. In this study, Sunagar et al. also evaluated the safety and immunogenicity of heterologous prime-boost vaccination with IM COVAXIN-prime followed by BBV154 IN administration which showed an acceptable reactogenicity profile comparable to the homologous COVAXIN/COVAXIN or BBV154/BBV154 vaccination. Heterologous vaccination of COVAXIN-prime and BBV154 booster also elicited superior and Omicron variant-specific protective immune responses compared with the homologous COVAXIN/COVAXIN immunization [155].

In a novel approach, Ishii, H. et al. investigated the efficacy of an IN vaccine expressing viral non-spike antigens against a SARS-CoV-2 challenge in cynomolgus macaques. Vaccinated macaques exhibited significantly reduced viral load in nasopharyngeal swabs post-challenge compared with unvaccinated controls. The viral control in the absence of SARS-CoV-2-specific nAbs was significantly correlated with vaccine-induced, viral-antigen-specific CD8+ T-cell responses. These results indicate that IN vaccination-induced CD8+ T cells can result in nAb-independent control of SARS-CoV-2 infection, highlighting the potential of vaccine-induced CD8+ T-cell responses in containing COVID-19 [92].

A recombinant trimeric spike protein adjuvanted with CpG oligonucleotides, ODN2006, was evaluated in a mouse model. The IN immunization successfully induced not only systemic spike-specific IgG Abs but also secretory IgA Abs in the nasal mucosa. Secretory IgA Abs showed a high protective ability against SARS-CoV-2 variants (Alpha, Beta, and Gamma variants) compared to IgG Abs in the serum. The nasal vaccine of this formulation induced a high number of IFN-γ-secreting cells in the draining cervical lymph nodes and a lower spike-specific IgG1/IgG2a ratio compared to that of subcutaneous vaccination with alum as a typical Th2 adjuvant [154].

#### 4.2.2. Clinical Trials

As of 3 March 2023, over a dozen mucosally delivered vaccines are being investigated in clinical trials (https://www.who.int/publications/m/item/draft-landscape-of-covid-19-candidate-vaccines accessed on 3 March 2023). These vaccines are based on viral vectors (replicating and non-replicating), live-attenuated viruses, nucleic acid, and recombinant proteins (Table 2). The interim immunogenicity and safety of an IN adenoviral-vectored COVID-19 vaccine (BBV154) in healthy adults compared with a licensed IM vaccine (Covaxin^®^) showed two IN doses of BBV154 were well tolerated with no safety concerns while eliciting superior humoral and mucosal immune responses compared to two IM Covaxin injections [161], whereas the phase-1 and phase-2 trials of an IN live-attenuated influenza virus vector-based vaccine (dNS1-RBD) demonstrated two doses of dNS1-RBD IN-spray induced weak humoral and CMI as well as mucosal immune responses against SARS-CoV-2 [162].

The preliminary report of the safety, tolerability, and immunogenicity of an aerosolized adenovirus type-5 vector-based COVID-19 vaccine (Ad5-nCoV) elicited neutralizing Ab responses and excellent safety with two doses of aerosolized Ad5-nCoV. The Ab response was comparable to one dose of intramuscular injection. In addition, both IN and IM routes induced a SARS-CoV-2 spike-specific polyfunctional Th1 response [163].

A phase I/II clinical trial was conducted by the Centre for Genetic Engineering and Biotechnology, Cuba, of a recombinant subunit vaccine CIGB-669. (CIGB-669) comprises RBD protein and HBV nucleocapsid (N) antigen (AgnHB, an immunopotentiator). The phase I/II studies involve groups that are given the CIGB-669 via IN and IM routes and schedules (RPCEC00000345).

Laboratorio Avi-Mex is conducting a phase II trial using the live Newcastle disease viral vector (rNDV)-based COVID-19 (AVX/COVID-12) vaccine via single dose administration with two different (IM or IN) routes in 396 healthy subjects (NCT05205746). Another IN vaccine, a live-attenuated vaccine against the respiratory syncytial virus (RSV) that is expressing the spike (S) protein of SARS-CoV-2 (MV-014-212) is under phase I trial by Meissa Vaccines, Inc. The MV-014-212 is administered as drops or a spray in the nose of healthy adults who are seronegative to SARS-CoV-2 and have not received a prior vaccine against COVID-19 (NCT04798001).

Finally, Bharat Biotech evaluated a replication-deficient chimpanzee ad-vectored vaccine (BBV154) in a phase III randomized multicenter clinical trial involving 3000 participants wherein participants were given two doses of vaccine via the IN route (CTRI/2022/02/040065). Recently, following the successful completion of the clinical trial, BBV154 has received restricted emergency use approval (in those aged above 18 years) from the Drugs Controller General of India (https://www.bharatbiotech.com/images/incovacc/incovacc-factsheet.pdf accessed on 10 March 2023). This success will hopefully spur additional research and development activities toward the development of novel IN vaccines against SARS-CoV-2.

While most vaccines currently under clinical trials target the IN route, a few have also targeted the oral route. VXA-CoV2-1 is a nonreplicating adenoviral-based vaccine expressing SARS-C-V2 antigen and dsRNA adjuvant. A phase 2 clinical trial of this vaccine is currently underway (NCT05067933). Another oral vaccine bacTRL-Spike is being investigated in a phase 1 clinical trial, which is based on *Bifidobacterium longum* (NCT04334980). This bacterium has been engineered to deliver plasmids encoding spike protein from SARS-CoV-2. The safety of two doses of orally administered CoV2-OGEN1 is being investigated in healthy subjects (NCT04893512) [164].

### 4.3. Systemic Prime-Mucosal Boost Strategy

It is important to note that previous studies have shown that the effectiveness of COVID-19 vaccines can vary depending on the route of administration. For example, a phase I study of the hAd5-based IN vaccine AdCOVID developed by Altimmune, Inc. showed lower immune response compared to other COVID-19 vaccines, leading to the discontinuation of further development of AdCOVID studies (https://ir.altimmune.com/news-releases/news-release-details/altimmune-announces-update-adcovidtm-phase-1-clinical-trial, accessed on 10 March 2023). Similarly, a phase 1 trial of IN vaccination with ChAdOx1nCoV-19 or live recombinant Newcastle disease virus-based COVID-19 vaccine (Patria) or live-attenuated influenza virus vector-based SARS-CoV-2 vaccine (dNS1-RBD) in healthy adults demonstrated an acceptable tolerability profile but induced neither a consistent mucosal Ab response nor a strong systemic response, suggesting that effective vaccination strategies should not be limited to a single route of administration [162,165,166].

Instead, a combination approach of parenteral prime and mucosal boost strategies may be more effective in balancing local and systemic immunity. This approach involves using an initial injection followed by a mucosal booster vaccination to elicit a better immune response, including the induction of IgA response and tissue-resident memory cells in the respiratory tract [122,167,168]. Studies have shown that two doses of Ad5-nCoV aerosolization induced nAb responses, similar to one dose of IM injection of Ad5-nCoV [163]. Furthermore, an aerosolized booster vaccination in Ad5-nCoV IM-primed recipients elicited strong IgG and nAb responses, suggesting that a systemic prime-intranasal boost can augment the immune response compared to an IN/IN homologous regimen [163]. Overall, choosing the right combination of administration routes can significantly improve the effectiveness of COVID-19 vaccines.

Two more COVID-19 vaccines are being evaluated under heterologous prime-boost regimens. Bharat Biotech completed a phase 2 interchangeability study to evaluate the immunogenicity and safety of COVAXIN^®^ with BBV154 (Adenoviral Intranasal COVID-19 vaccine) in healthy volunteers (CTRI/2021/09/036257). Both the vaccines COVAXIN and BBV154 are proved to be safe in heterologous prime-boost regimens and immunogenic in all combinations. The immunogenicity and safety of the BBV154 IN vaccine were further evaluated and compared with licensed injectable COVID-19 vaccines in a phase 3 clinical trial in individuals primed with COVAXIN or COVISHIELD (NCT05567471).

Similarly, CanSino Biological Inc./Beijing Institute of Biotechnology, completed the phase I/II trial with hAd5-nCoV as a stand-alone IN vaccine (NCT04840992) following which they evaluated the safety and immunogenicity of the heterologous prime-boost immunization of hAd5-nCoV vaccine via aerosol (Ad5-nCoV-IH) or IM (Ad5-nCoV-IM) in 360 individuals, who were vaccinated (primed) with three doses of an inactivated COVID-19 vaccine (CoronaVac) more than 6 months ago (NCT05303584).

These studies indicate the increased interest in adopting a systemic prime and mucosal boost as a novel strategy for better control of the SARS-CoV-2 infection.

### 4.4. Adjuvants for Mucosal Vaccines against COVID-19

Inducing an immune response on the mucosa typically requires a higher dose of antigen compared to parenteral immunization, due to the dilution of the vaccine in mucus and partial excretion through ciliary movements and mucus in the airways [169]. Moreover, overcoming physical and biochemical obstacles, including acidic pH and enzymes, is crucial to achieving successful mucosal immunization [170]. Adjuvants can enhance or modulate the humoral or cellular immune response to the presented antigen, prevent the body’s tolerogenic responses to the antigen, recruit and activate APCs, and engage immune cell populations in mucosal tissue such as innate lymphoid cells and natural killer T cells [131,171,172,173,174,175,176,177,178]. However, only a few effective and safe adjuvants have been identified so far, including those targeting M cells, dendritic cells, and activating invariant natural killer T cells, which offer promising prospects for improving mucosal immunization [179].

In preclinical studies, it has been found that immunization of the lungs with liquid and dry vaccines induces both systemic and mucosal immune responses. However, the response in terms of mucosal IgA Abs is weak, and the use of adjuvants is necessary to obtain effective immunization by the mucosal route [134,180]. Combining selected antigens with appropriate adjuvants to prepare mucosal vaccines is a simple process that can significantly enhance the immune response while reducing the required dose of antigen [181]. While the majority of approved adjuvants have been studied in the context of their use in conventional vaccines, it is not fully understood how they act in mucosal immune responses. Cholera toxin (CT) and *Escherichia coli* heat-labile enterotoxin (LT) are the best-known mucosal adjuvants, which interact with the surface of dendritic cells and enhance the induction of B-cell clones [182,183]. Various polymers such as chitosan and liposomes are used as adjuvanted carriers [184], while ISCOMs [185] have also been found to be effective adjuvants for mucosal vaccines. In addition, specialized molecules such as lectins facilitate the targeting of the antigen to surface markers of epithelial and dendritic cells, which increases the efficiency of antigen uptake by APCs [186].

Aluminum salts are a commonly used adjuvant in human vaccines against pathogens such as human papilloma virus, hepatitis A and B viruses, influenza type B, tetanus, or diphtheria [187]. They have been tested in potential preparations against COVID-19 and found to generate a strong humoral response in preclinical studies [188,189]. However, aluminum compounds used as adjuvants have limitations, such as poor immuno-stimulation of cellular immune responses and limited efficacy against intracellular pathogens. Additionally, they are not effective in inducing mechanisms aimed at the activation and recruitment of B and T cells in mucosal tissues [190]. To overcome these limitations, there is an urgent need to develop novel adjuvants that enhance the immunogenicity of antigens without causing toxicity. These adjuvants should be universal for many antigens and effective in reducing the effective doses of vaccines and improving long-term stimulation of the systemic as well as mucosal immune response including antigen-specific humoral, CMI—specifically tissue-resident T cells. It is essential to choose the appropriate adjuvant to maximize the effectiveness of a potential vaccine [191,192].

### 4.5. Adjuvant-Free Approach to Mucosal Vaccination

APCs have long been a target for vaccine development due to their central role in antigen processing and presentation [193,194,195,196,197,198,199,200,201,202]. FcγRI is an activating receptor with an immunoreceptor tyrosine-based activation motif (ITAM) in its cytoplasmic domain and is expressed exclusively on macrophages and DCs, which makes it uniquely suitable for APC targeting [203,204,205]. Numerous studies have demonstrated that targeting antigens to FcγRI in vitro and in vivo can enhance humoral and cell-mediated immune responses [206,207,208,209,210,211,212,213,214,215,216,217].

In two separate studies, the hFcγRI-targeting approach was evaluated in a mouse model of pneumococcal pneumonia [218,219]. In both studies, APC targeting was realized with a monoclonal Ab that binds to hFcγRI (α-hFcγRI) [220]. When PspA, a Streptococcus pneumoniae (Sp) surface antigen, was fused to the α-hFcγRI, the APCs acquire, process, and present the antigen more efficiently (Figure 3) and as a result, the immunogenicity and efficacy of the antigen is significantly increased. Importantly, the α-hFcγRI recognizes the hFcγRI via an epitope separate from the Fc-binding domain of the receptor, which allows it to retain binding even when the host Abs such as IgG occupies the Fc-binding domain. Upon IN immunization, the α-hFcγRI-PspA (fusion protein/FP)-induced Ab response in the serum as well as in the respiratory mucosa. The mucosal Ab response included IgG and IgA. In addition, we also observed Th17 and Th22 responses in the mouse lungs following IN immunization (Figure 4). The immunized mice were protected against lethal pneumococcal pneumonia [219]. We also observed that IN immunization induced significantly higher IgG and IgA responses in the BAL compared to IM immunization (unpublished data). This approach can be adapted for mucosal vaccines against SARS-CoV-2 utilizing a genetic fusion of spike protein or the RBD alone as target antigens. These antigens may be genetically fused to the α-hFcγRI Ab and expressed recombinantly. We anticipate that IN immunization of α-hFcγRI-Spike/α-hFcγRI-RBD fusion proteins will induce nAbs in blood as well as in the respiratory mucosa. In addition, lung resident T-cell response will also be elicited, which in combination with Abs will contribute to robust control of SARS-CoV-2 infection.

In addition to targeting protein antigens, we have also developed a novel approach to target the whole-cell antigens with α-hFcγRI to the antigen-presenting cells, via IN immunization. Mannose-binding lectins can associate with viral and bacterial cells by binding to various sugar moieties attached to the surface glycoproteins. We have constructed a fusion protein of α-hFcγRI with human MBL. This fusion protein (α-hFcγRI-MBL) binds efficiently to the formaldehyde-inactivated *Francisella tularensis* live vaccine strain. We evaluated the efficacy of the inactivated *F. tularensis* (iFt)- MBL-α-hFcγRI complex in a mouse model by IN immunization. The iFt- MBL-α-hFcγRI complex induced significantly higher Ab response and protection against a lethal pulmonary *F. tularensis* challenge compared to iFt alone (manuscript in preparation). MBL can efficiently bind to various microbial cells by recognizing a variety of sugar moieties present on microbial surfaces. Accordingly, we evaluated the binding of MBL-α-hFcγRI to a number of microbes including Sp, *Klebsiella pneumoniae*, and *Streptococcus aureus* (manuscript in preparation). As expected, the MBL-α-hFcγRI did bind efficiently to all these microbes. Recently, we have observed that the α-hFcγRI-MBL fusion protein binds efficiently to SARS-CoV-2 as well (Figure 5), suggesting its applicability in vaccine development against SARS-CoV-2. It is important to note that inactivated SARS-CoV-2 has been evaluated in multiple clinical trials as a vaccine, and some of these have also been approved for human use. Thus, we anticipate that a complex of (inactivated) SARS-CoV-2- MBL-α-hFcγRI can generate protective immunity and upon IN immunization can also elicit robust immunity in respiratory mucosa against SARS-CoV-2.

## 5. Conclusions

Current vaccines significantly reduce the severity of SARS-CoV-2 infections but remain ineffective against novel VOCs. In addition, the weakening of vaccine-induced immunity is associated with breakthrough infections among vaccinees. On the other hand, several lines of evidence suggest that mucosal immunization via natural infection or vaccination induces a more robust immune response in respiratory mucosa, which is the prime target of SARS-CoV-2 infection. In this regard, vaccine approaches such as systemic prime and mucosal booster and mucosal vaccination offer better alternatives to mitigate COVID-19. It is important to note that the rapid control of the pandemic can be achieved through immunity, which limits the person-to-person spread (or transmission) of the pathogen. Since the transmission occurs via the respiratory route, pathogen-specific immunity in respiratory mucosa can better resist the infection, limiting the community spread. Moreover, novel tools to more accurately evaluate T cell-mediated immunity and mucosal immunity are urgently required. In addition, further studies are needed to develop safe and effective mucosal vaccines against SARS-CoV-2.

## Figures and Tables

**Figure 1 vaccines-11-00849-f001:**
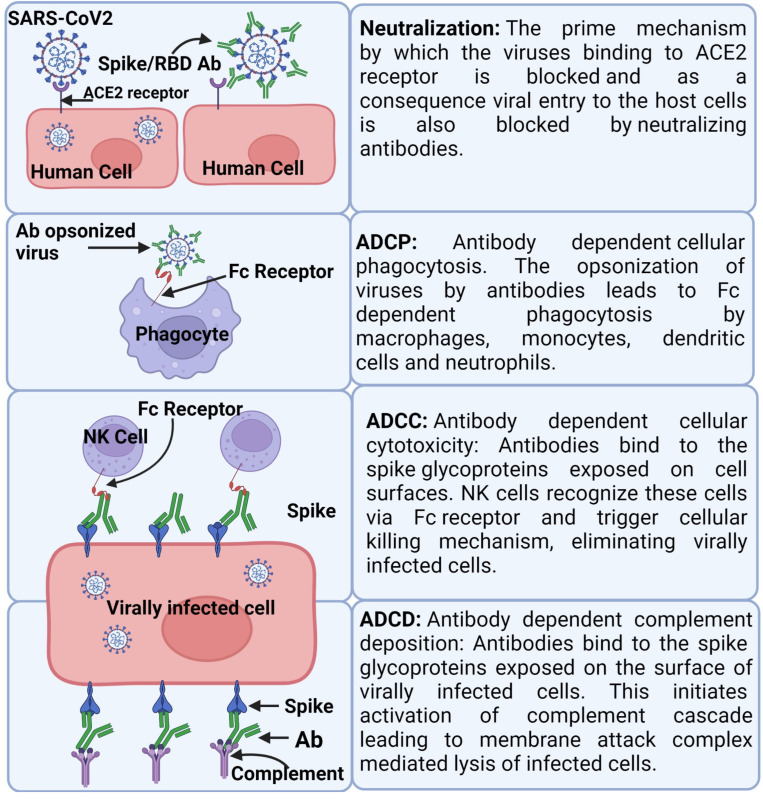
Antibody mediated protection against SARS-CoV-2: Created with BioRender.com (https://www.biorender.com/ accessed on 3 March 2023).

**Figure 2 vaccines-11-00849-f002:**
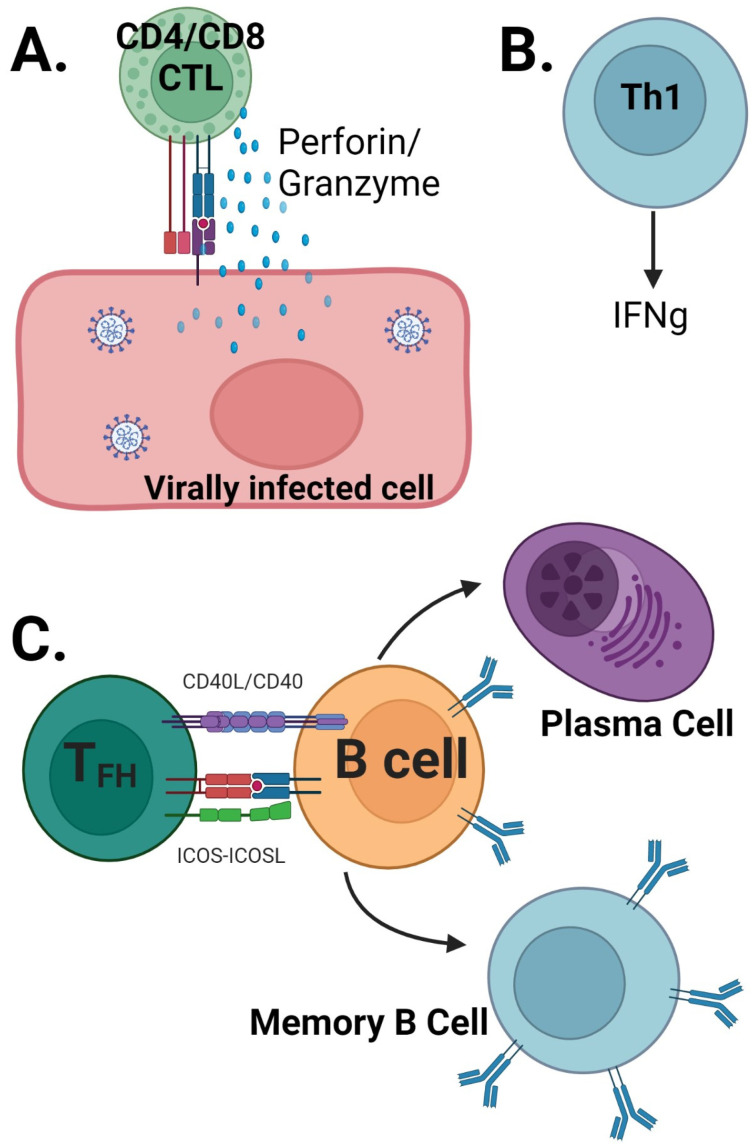
T-cell mechanism of protection against SARS-CoV-2: CD4/CD8 CTL recognize virally infected cells via MHC-peptide TCR interaction, which leads to killing of the virally infected cells (**A**). Th1 cell-secreted cytokines mediate antiviral effect (**B**). T_FH_ cells help in the generation and maintenance of antibody response by helping B cells (**C**).

**Figure 3 vaccines-11-00849-f003:**
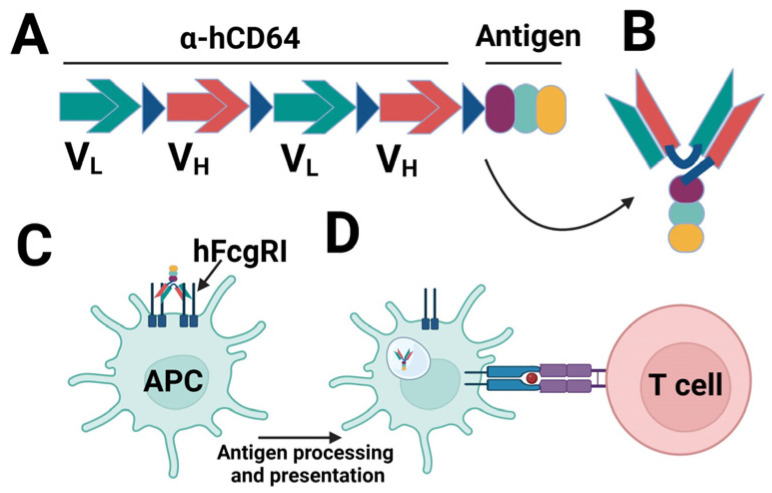
α-hCD64-targeted vaccine design: (**A**) The map of the candidate vaccines. (**B**) Folding of the α-hCD64 and antigen moieties. (**C**) The α-hCD64-Antigen binds to the hFcγRI on APCs. (**D**) APCs in turn process and present the antigens to the naïve T cells.

**Figure 4 vaccines-11-00849-f004:**
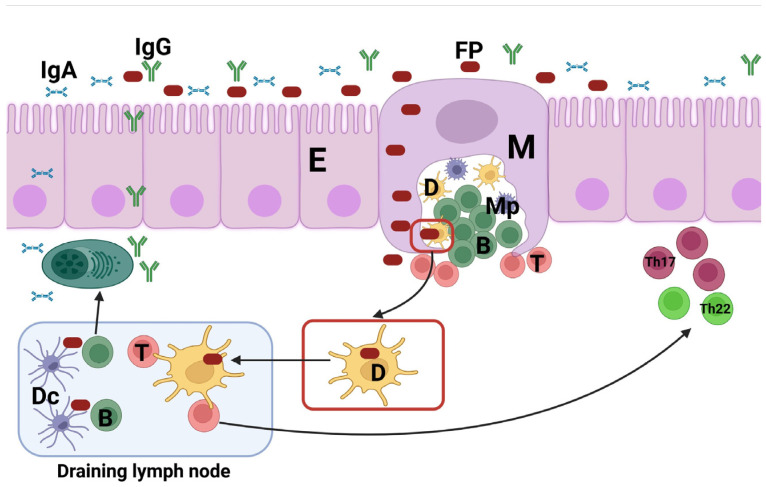
Immune response elicited by hFcgRI-targeted vaccines. Following intranasal immunization, the fusion proteins (FPs) containing the APC-targeting component α-hCD64 and the antigen crosses through M cells (M) at the nasal-associated lymphoid tissues (NALT). Following transcytosis, the FPs are taken up by antigen-presenting cells including DCs (D) and macrophages (Mp). APCs can either present the antigens to the T cells (T) at the NALT or at the draining lymph nodes, ensuing adaptive immune response including the generation of antigen-specific IgG, IgA, and Th17 and Th22 responses. The IgG and IgA at the mucosa are secreted (Transcytosis: arrow across the epithelium) to the mucosa thereby preventing the invasion of the cognate pathogen. E: Epithelial cells, Dc: follicular dendritic cells.

**Figure 5 vaccines-11-00849-f005:**
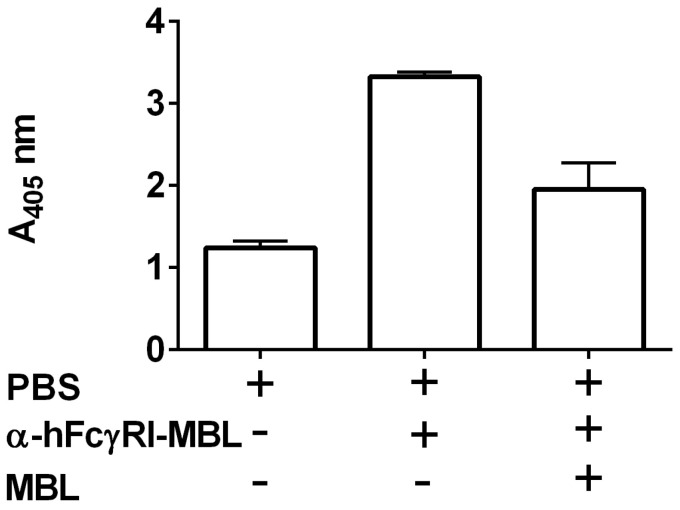
Binding of α-hFcγRI-MBL with inactivated SARS-CoV-2: ELISA plates were coated with inactivated SARS-CoV-2 overnight and blocked with 5% BSA. PBS, α-hFcγRI-MBL, and MBL were added in appropriate wells and incubated for one hour at 4C. The binding of α-hFcγRI-MBL was evaluated by sequential treatment with anti-MBL antibody and alkaline phosphatase-conjugated secondary antibody. Result: In comparison with PBS, the increase in ELISA signal with α-hFcγRI-MBL indicates binding of the fusion protein to inactivated SARS-CoV-2, while the reduction in signal in the presence of MBL signifies specificity of binding.

**Table 2 vaccines-11-00849-t002:** Mucosal SARS-CoV-2 Vaccines in Clinical trials.

Vaccine Platform	Type	Vaccine	Developer	Antigen	Route	Clinical Trial
Viral Vector (Non-replicating)	Simian adenovirus vector	BBV154	Bharat Biotech	Prefusion Spike	Intranasal (IN)	CTRI/2022/02/040065 (phase 3)
		ChAdOx1-S	University of Oxford	Spike	IN	NCT04816019 (phase 1)
	Human adenovirus type 5 vector	AdCOVID	Altimmune	Receptor-binding domain (RBD)	IN	NCT04679909 (phase 1) (Discontinued)
		Ad5-nCoV-IH	CanSino Biological Inc.	Spike	Inhaler (IH)	NCT05303584 (phase 4)
		VXA-CoV2-1 Ad5	Vaxart	Spike	Oral	NCT05067933 (phase 2)
	Parainfluenza virus	CVXGA1	CyanVac LLC	Spike	IN	NCT04954287 (phase 1)
	Human or simian adenovirus vector	Ad5-triCoV/Mac or ChAd-triCoV/Mac,	McMaster University	Nucleocapsid, spike and RNA polymerase proteins	Aerosol (AE)	NCT05094609 (phase 1)
	Modified vaccinia virus Ankara	MVA-SARS-2-ST Vaccine	Hannover Medical School	Spike	IH	NCT05226390 (phase 1)
Viral vector (Replicating)	Live-attenuated Influenza virus	DelNS1-nCoVRBD LAIV	University of Hong Kong	RBD	IN	ChiCTR2100051391 (phase 3)
	Live-attenuated Respiratory syncytial virus	MV-014-212	Meissa Vaccines, Inc.	Spike	IN	NCT04798001 (phase 1)
	Live recombinant Newcastle Disease Virus (rNDV) vector vaccine	NDV-HXP-S	Laboratorio Avi-Mex	Prefusion spike	IN and IM	NCT05205746 (phase 2)
Nucleic acids based	DNA vaccine	bacTRL-Spike	Symvivo Corporation	Spike	Oral	NCT04334980 (phase 1)
Live-attenuated virus	Live-attenuated SARS-CoV-2	COVI-VAC	Codagenix/Serum Institute of India	Live-attenuated SARS-CoV-2	IN	ISRCTN15779782 (phase 3)
Recombinant	Protein subunit	CIGB-669	Center for Genetic Engineering and Biotechnology, Cuba	Protein subunit AgnHB (RBD)	IN	RPCEC00000345 (phase 1/2)
		Razi Cov Pars	Razi Vaccine and Serum Research Institute	Spike	IN and IM	IRCT20210206050259N3 (phase 3)
		CoV2-OGEN1,	USSF/Vaxform	RBD	Oral	NCT04893512 (phase 1)

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
