# Peer review of "SARS-CoV-2: Immunity, Challenges with Current Vaccines, and a Novel Perspective on Mucosal Vaccines"

_vaccines, 2023, doi:10.3390/vaccines11040849_

Round 1

Reviewer 1 Report

The proposed review discusses the immunity induced by SARS-CoV-2 vaccines and the prospects for the development of mucosal vaccines. I very much enjoyed reading this manuscript which is clear and well-constructed. It provides an exhaustive state of the art on nasal vaccination which is a central public health challenge. 

I have some suggestions of improvements to submit to the authors.

- Several typos or repetitions need to be corrected (for example lines 59, 64 (nAbs already abbreviated), 78 (ref 29), p10 L25, p15 L298/300/302, p18 L82, p20 L146)

- Explain BAL (p11), ITAM and FP (p18), ScFv (figure 4), iFt and Sp (p19)

- The figure 1 is not cited in the text.

- Figure 2: insertion of A,B,C,D would make the figure easier to read.

- Figure 4: arrows indicating the secretion of Abs through the epithelium would be appreciated. Besides, E and M are not explained in the legend (epithelium and M cells?) and the abbreviation of NALT should be indicated line 103. Dendritic cells are named D in the mucosa and then Dc in the LN.

- The review especially focuses on nasal administration rather that mucosal in general as this administration route is the more studied. Thus, it would be more appropriate to indicate ‘nasal vaccines’ in the title, or discuss about other mucosal route, such as oral route that is mentioned in table 2 but not commented at all.

- p4 and 5: “Moreover, measuring antigen-specific T cells is more technically challenging and expensive than measuring Ab responses “ “However, it should be noted that the rhesus monkey model is a challenging model to observe T cell protective immunity”. Why? Maybe an explanation of sampling that would have to be performed for Tcell activity measurements would help to understand the evoked hurdles, compared to ‘simple’ sampling of peripheral blood.

- p3 “2.1.4 Mechanisms of Ab induction” and p6: “2.2.3 T cell mechanisms of protection” are introductions to cellular pathways and would be appreciated at the beginning of each respective parts.

- Some references are missing: p3 L103, p11 L81

- p12: “4.3. Heterologous prime-boost vaccination” is an interesting paragraph but as no mention of mucosal immunity, as expected from the title of Part 4 “Perspectives on mucosal vaccine strategies”.

- p13: “4.4. Mucosal vaccines” in my opinion this part should be 4.2 as it introduces the concepts of IN vaccines. “Systemic prime and mucosal boost” could come after this part as an evolution of the homologous vaccination scheme.

Author Response

Reviewer 1

General: Thank you very much for taking the time to review our manuscript and for the detailed evaluation, suggestions and comments. This will certainly improve the quality of our manuscript.

Comments and response (Re:)

- Several typos or repetitions need to be corrected (for example lines 59, 64 (nAbs already abbreviated), 78 (ref 29), p10 L25, p15 L298/300/302, p18 L82, p20 L146).

Re: We have made corrections as advised.

- Explain BAL (p11), ITAM and FP (p18), ScFv (figure 4), iFt and Sp (p19)

Re: We have made corrections as advised.

- The figure 1 is not cited in the text.

Re: We have added Fig 1 in the text (p5, line 122).

- Figure 2: insertion of A,B,C,D would make the figure easier to read.

Re: We have made corrections as advised (p 5, Fig 2).

- Figure 4: arrows indicating the secretion of Abs through the epithelium would be appreciated. Besides, E and M are not explained in the legend (epithelium and M cells?) and the abbreviation of NALT should be indicated line 103. Dendritic cells are named D in the mucosa and then Dc in the LN.

Re: We have made corrections as advised. Dc in the Ln is actually follicular dendritic cells. We have indicated it in the revised figure legends (p24, Fig 4).

- The review especially focuses on nasal administration rather that mucosal in general as this administration route is the more studied. Thus, it would be more appropriate to indicate ‘nasal vaccines’ in the title, or discuss about other mucosal route, such as oral route that is mentioned in table 2 but not commented at all.

Re:  We have added a paragraph detailing the oral vaccines that are currently being investigated (p20, line 28-35).

- p4 and 5: “Moreover, measuring antigen-specific T cells is more technically challenging and expensive than measuring Ab responses “ “However, it should be noted that the rhesus monkey model is a challenging model to observe T cell protective immunity”. Why? Maybe an explanation of sampling that would have to be performed for Tcell activity measurements would help to understand the evoked hurdles, compared to ‘simple’ sampling of peripheral blood.

Re: We have added explanation of this point (p6, line 158-160 and p7 line 236-237).

- p3 “2.1.4 Mechanisms of Ab induction” and p6: “2.2.3 T cell mechanisms of protection” are introductions to cellular pathways and would be appreciated at the beginning of each respective parts.

Re: We have made corrections as advised (p3, line 101 and p8 line 255-256).

- Some references are missing: p3 L103, p11 L81

Re: We have added references as advised (ref 39, 40 and 141, 144, 146, 149 and 150).

- p12: “4.3. Heterologous prime-boost vaccination” is an interesting paragraph but as no mention of mucosal immunity, as expected from the title of Part 4 “Perspectives on mucosal vaccine strategies”.

Re: Thank you for pointing that out. We have removed this section from the manuscript, and we agree that removing this part does not affect the central thesis of this manuscript.   

- p13: “4.4. Mucosal vaccines” in my opinion this part should be 4.2 as it introduces the concepts of IN vaccines. “Systemic prime and mucosal boost” could come after this part as an evolution of the homologous vaccination scheme.

Re: Thank you for pointing that out. We have now moved the mucosal vaccines section as 4.2. and  Systemic prime and mucosal boost” as 4.3. 

Reviewer 2 Report

This is an interesting study on COVID-19 vaccine. Please note the
following point:

1.  the title of 3.3 is “Mucosal immunity induced by parenteral”, but
this section does not mention parenteral vaccination. Should we consider
changing the title? Or add more content and references about it.

2. The authors refer to antibody dynamics after vaccination. Would you
consider a reference to illustrate this point? DOI: 10.3390/vaccines10050647

3. Please pay attention to standard writing. The short line in the word
drops the &ldquo

Author Response

Reviewer 2

General: Thank you very much for taking the time to review our manuscript and for the detailed evaluation, suggestions and comments. This will certainly improve the quality of our manuscript.

Comments and response (Re: )

  1. the title of 3.3 is “Mucosal immunity induced by parenteral”, but
    this section does not mention parenteral vaccination. Should we consider
    changing the title? Or add more content and references about it.

Re: Thank you for the comment. This part discusses about the intramuscular (IM) vaccines which is a parenteral route.

  1. The authors refer to antibody dynamics after vaccination. Would you
    consider a reference to illustrate this point? DOI: 10.3390/vaccines10050647

Re: Thank you for the suggestion. We have added this reference (Ref number 115) in section 3.1.

  1. Please pay attention to standard writing. The short line in the word
    drops the &ldquo

Re: Thank you for the suggestion. We have adhered to the standard writing while writing this manuscript.

Reviewer 3 Report

The manuscript is easy and nice to read. The topics covered in the review are adequate and complete. The order in which each topic is described is appropriate, making reading very enjoyable. An adequate number of references have been selected for each topic and their quality is very good; most of the cited authors are referents of each area. The authors present, at the end of the manuscript, some of their own results (published and unpublished data) obtained using two different adjuvant free vaccines to elicit systemic and mucosal immunity against pulmonary pneumococcal infection or lethal pulmonary Francisella tularensis. The authors hypothesize that similar approaches would induce systemic and mucosal immune response against SARS-CoV-2.

Minor spelling mistakes:

Table I Sinopharm

Page 10, line 15 shows

Ref #145 is not correct (Please check all the References).

Author Response

Reviewer 3

General: Thank you very much for taking the time to review our manuscript and for the detailed evaluation, suggestions and comments. This will certainly improve the quality of our manuscript.

Comments and response (Re: )

--Table I Sinopharm

Re: Thank you for the suggestion. We have made corrections as advised.

--Page 10, line 15 shows

Re: Thank you for the suggestion. We have made corrections as advised.

--Ref #145 is not correct (Please check all the References).

Re: Thank you for pointing that out. We have removed the reference and verified all the refences.

Reviewer 4 Report

The vaccine used for COVID-19 contributed greatly to reducing the spread of the pandemic as well as the severity of the disease, hospitalizations and deaths around the planet. We can therefore suggest that other compartments of the immune system must be activated to improve the efficacy of existing or new vaccines. Yet despite all the efforts, we must keep in mind that this is an ongoing emergency, as new variants emerge with high frequency. To circumvent this, other strategies must be applied. The review covers different aspects and suggests others. Mucosal immunity is important but we can still consider it preliminary. More data is needed for the future. Who is to say that the two will not have to act together! Everything is intended to improve. The review is very well done.  I would like to suggest to revise some points and draw definite conclusions of the  review !.  Many important aspects were very well discussed in the review !

Revise the abstract !

Author Response

Reviewer 4

General: Thank you very much for taking the time to review our manuscript and for the detailed evaluation, suggestions and comments. This will certainly improve the quality of our manuscript.

Comment and response (Re:)

Revise the abstract !

Re: We have revised the abstract as advised!